# Green Synthesis of Silver Nanoparticles Using Pomegranate and Orange Peel Extracts and Their Antifungal Activity against *Alternaria solani*, the Causal Agent of Early Blight Disease of Tomato

**DOI:** 10.3390/plants10112363

**Published:** 2021-11-02

**Authors:** Yasser S. Mostafa, Saad A. Alamri, Sulaiman A. Alrumman, Mohamed Hashem, Zakaria A. Baka

**Affiliations:** 1Department of Biology, College of Science, King Khalid University, Abha P.O. Box 9004, Saudi Arabia; ysolhasa1969@hotmail.com (Y.S.M.); amri555@yahoo.com (S.A.A.); salrumman@kku.edu.sa (S.A.A.); drmhashem69@yahoo.com (M.H.); 2Department of Botany and Microbiology, Faculty of Science, Assiut University, Assiut P.O. Box 71515, Egypt; 3Department of Botany and Microbiology, Faculty of Science, Damietta University, New Damietta P.O. Box 34517, Egypt

**Keywords:** *Alternaria solani*, silver nanoparticles, pomegranate peel, orange peel, early blight, tomato

## Abstract

This study aimed to synthesize silver nanoparticles (AgNPs) by pomegranate and orange peel extracts using a low concentration of AgNO_3_ solution to controlearly blight of tomato caused by *Alternaria solani*. The pathogen was isolated from infected tomato plants growing in different areas of Saudi Arabia. The isolates of this pathogen were morphologically and molecularly identified. Extracts from peels of pomegranate and orange fruits effectively developed a simple, quick, eco-friendly and economical method through a synthesis of AgNPs as antifungal agents against *A. solani*. Phenolic content in the pomegranate peel extract was greater than orange peel extract. Phenolic compounds showed a variation of both peel extracts as identified and quantified by High-Performance Liquid Chromatography. The phenolic composition displayed variability as the pomegranate peel extract exhibited an exorbitant amount of Quercitrin (23.62 mg/g DW), while orange peel extract recorded a high amount of Chlorogenic acid (5.92 mg/g DW). Biosynthesized AgNPs were characterized using UV- visible spectroscopy which recorded an average wavelength of 437 nm and 450 nm for pomegranate and orange peels, respectively. Fourier-transform infrared spectroscopy exhibited 32x73.24, 2223.71, 2047.29 and 1972.46 cm^−1^, and 3260.70, 1634.62, 1376.62 and 1243.76 cm^−1^ for pomegranate and orange peels, respectively. Transmission electron microscopy showed spherical shape of nanoparticles. Zetasizer analysis presented negative charge values; −16.9 and −19.5 mV with average particle sizes 8 and 14 nm fin case of pomegranate and orange peels, respectively. In vitro, antifungal assay was done to estimate the possibility of biosynthesized AgNPs and crude extracts of fruit peels to reduce the mycelial growth of *A. solani*. AgNPs displayed more fungal mycelial inhibition than crude extracts of two peels and AgNO_3_. We recommend the use of AgNPs synthesized from fruit peels for controlling fungal plant pathogens and may be applied broadly and safely in place by using the chemical fungicides, which display high toxicity for humans.

## 1. Introduction

Tomatoes (*Lycopersicon esculentum* Mill.), family Solanaceae considered as a very important vegetable in the world, including Saudi Arabia. Early blight, started by *Alternaria solani* (Ell. and Mart.) Jones and Grout. is well-known foliar disease that causes yield losses equal to 70% [1]. Management of this disease depends on several uses of synthesized fungicides through late stages of tomato production [2]. By using chemical pesticides to control, plant diseases are limited due to their toxicity, prolonged periods of degradation and pollution of the environment.

Between these options are biological plant products, which are bio-efficacious, biodegradable, cost-effective and safe for the environment. Many investigators used biological plant products to control plant pathogens [3,4,5,6,7,8,9,10,11,12,13,14,15].

A phytochemical from medicinal plants display antimicrobial actions have the potential to achieve this need because their structures vary from those of the more studied microbial sources and hence their mode of action may too likely differ. Recently, special care has been concentrated on the wastes developing from the industrial processing of fruits as an outstanding source of phenolic compounds. Investigation is focused on finding ways to use by-products in medicinal plants, owing to their health and technological benefits [16]. Extracts gained from fruit peels are a mixture of phenolic compounds and exhibit antimicrobial effects [17]. By using a solvent extraction system, antimicrobial compounds can be formed from these peels. Usually, fruit peels are discarded like garbage and a significant amount of such wastes are formed during industrial processing. These economic sources for gaining antimicrobials can be used in the avoidance of diseases caused by plant pathogenic microbes.

The increasing interest in the quantities of organic household waste generated all around the world is creation research emphasis on the recovery of food residues. Many fruit origin household wastes, like fruit peels, are not frequently eaten, and are often richer than the edible tissues in different ingredients as phenolics, meaning that many bioactive chemicals could be extracted from them [18].

In recent years, nanotechnology has been established as an important field of modern research as groundbreaking technology interdisciplinary with physics, chemistry, biology and medication [19]. Moreover, nanotechnology can be defined as the study and application of building units called nanoparticles (NPs) with dimensions smaller than 100 nm. Nanobiotechnology is the branch that merges biological principles with physical and chemical procedures to produce NPs with specific functions [20].

Means of a biological source that are non-toxic, environmental, and low-cost technology for large-scale fabrication synthesized the silver nanoparticles (AgNPs). The synthesized AgNPs showed significant antifungal activity against many phytopathogenic fungi [21,22]. Furthermore, AgNPs are regularly expended in several treatments because of their exclusive optical possessions, excessive-constancy, and powerful conjugation capability with biomolecules [23]. Numerous chemical, photochemical, electrochemical and biological methods are involved in the synthesis of different sizes and shapes of the NPs to serve diverse purposes [24]. Chemical and physical approaches are conventionally concerned with NP synthesis that contaminates the environment. Hence, there is a necessity for “Green synthesis” of NPs that is hygienic, safe and environmentally friendly, and whichdoes not include high temperature, pressure, energy and poisonous chemicals [25]. Silver NPs have bee recognized as having an antifungal role for a long time, but their practices for controlling plant pathogens under field conditions has been limited because of the lack of information concerning their impact on plant and indigenous soil microorganisms. Moreover, certain descriptions designate antifungal and antibacterial actions of AgNPs in field and greenhouse situations [26]. Furthermore, exploitation in the route of AgNP synthesis, their structural properties and surface coating are being used for improving the antimicrobial action of silver NPs [27]. Silver NPs were also synthesized from numerous plant roots, seeds, leaves, stems, flowers and fruits for diverse purposes [28].

The aim of this investigation is to biosynthesize AgNPs using the extracts gained from the peels of pomegranate and orange fruits. Antifungal potential of these AgNPs against *A. solani*, the causal fungus of early blight of tomato plants, was also considered in this work.

## 2. Results

### 2.1. Identification of the Fungal Pathogen

#### 2.1.1. Morphological Identification of Alternaria Isolates

Eight fungal isolates of *A. solani* were obtained from field-infected tomato plant cultivar Super Strain B. These isolates were varied in their morphological characteristics. Conidia features varied significantly between the isolates, though they were all solitary. The observed morphological features characterized the isolates into four groups as recorded in Table 1. Group 2 was the largest comprising three isolates (A2, A3, A8) producing an irregularly shaped colony of greenish-brown color and characterized by a dark brown substrate color and gray border. These isolates had septate conidia with three transverse septa and elongated and unbranched beaks. Group 1 had one isolate (A1) characterized by a circular dark brown colony with a dark gray substrate color and gray margin. The member of this group had septate conidia with three transverse septa, one longitudinal septum and a branched slender short beak. Group 3 includes two isolates (A4, A6) alongside a circular dark gray colony having a greyish brown substrate and a white margin. Their conidia were septate with two transverse septa, one longitudinal septum and an elongated unbranched beak. Group 4 contained two isolates (A5, A7) with a circular colony of gray color, brown substrate and greyish white margin. These isolates had septate conidia with four transverse septa and an unbranched slender short beak.

#### 2.1.2. Molecular Characterization of Alternaria Isolates Using Sequence Analysis of the ITS Region

The PCR amplification of the ITS region of eight isolates of *A. solani* resulted in a product of about 570 bp. The band size did not vary between the fungal isolates in Figure 1. Means of the nucleotide BLAST program carried sequence similarity searches of the eight isolates out, National Center for Biotechnological Information (NCBI) at the U.S.A. All isolates were absolutely identified as *A. solani.* Four *A. solani* isolates (A1, A2, A3, A8) showed 100% nucleotide similarity to Gene Bank clone 107 with accession number MN871620. Two isolates (A4, A6) showed 99.50% nucleotide similarity to Gene Bank clone 180 with accession number MN871615. The last two isolates (A5, A7) showed 99.85% nucleotide similarity to Gene Bank clone 42 with accession number MN871610 (Table 2).

### 2.2. Pathogenicity Tests

The infected tomato plants by all *Alternaria solani* isolates caused symptoms of early blight with diverse levels of disease severity. The got data verify that isolates A1, A3 and A5 were extremely aggressive and produced the maximum disease severity. Isolates A2, A4, A7 and A8 showed the lowest disease severity on tomato plants. According to these results, isolate A1 was used in successive tests. Figure 2 showed the infected tomato plant by *A. solani* (Isolate A1) under greenhouse conditions.

### 2.3. Total Phenolic Content

Table 3 reveals the total phenolic in methanolic extracts of the peels of pomegranate and orange. The extract of pomegranate peel exhibited a high content of total phenolics than orange peel viz; 105.4 and 86.32 mg/g DW, respectively.

### 2.4. Quantitative Phenolic Compounds Present in Peel Extracts of Pomegranate and Orange Using High Performance Liquid Chromatography (HPLC)

Recognized phenolic compounds in the methanol extracts of pomegranate and orange peels were enumerated using HPLC are presented in Figure 3 and Figure 4. As obvious from the data in Figure 3 that Quercitrin recorded the highest amount (23.62 mg/g DW) in pomegranate peel, while Caffeic acid recorded the lowest value (3.48 mg/g DW). Thus, the relative contents of these composites exhibited inconsistency. Figure 4 recorded the highest amount of Chlorogenic acid (5.92 mg/g DW) in orange peel, while *P*-cumaric acid recorded the lowest value (0.09 mg/g DW).

### 2.5. Characteriuzation of AgPNs

#### 2.5.1. Ultraviolet–Visible Spectroscopy (UV-Vis Spectroscopy)

After the addition of AgNO_3_ to pomegranate and orange peel extracts, the color of the mixture changed from yellow to dim brown and from bright red to dim brown, respectively (Figure 5). Spectral analysis from UV-visible (UV-Vis) spectroscopy revealed that the solution peaked at an average wavelength of 437 nm and 450 nm for pomegranate and orange peels, respectively, after 72 h of reaction time, as presented in Figure 6.

#### 2.5.2. Fourier Transform Infrared Spectroscopy (FTIR)

Measurements by FTIR were delivered to explain and approve the probable development of bio-reduction and effectual steadying of biosynthesized AgNPs by means of extracts of pomegranate and orange peels. FTIR spectra recognized the reduction of mixtures in extracts. FTIR groups of pomegranate peel extract were conditional by 3273.24, 2223.71, 2047.29 and 1972.46 cm^−1^ (Figure 6A) and those of orange peel extract were inferred at 3260.70, 1634.62, 1376.62 and 1243.76 cm^−1^ (Figure 7).

#### 2.5.3. Transmission Electron Microscopy

As shown in Figure 8, optimized biosynthesized AgNPs were analyzed using TEM showed size range 10.0–16.69 nm for the extract of pomegranate peel and 9.45–17.15 nm for the extract of orange peel. The nanoparticles were spherical with very good distributions.

#### 2.5.4. Zeta Potential and Particle Size

The Zeta potential and size of the nanoparticle suspension were analyzed using the Zetasizer analysis and represented in Table 4 and Figure 9. The average particle size of pomegranate peel was 8 nm, while that of orange peel was 14 nm. Zeta potential results are shown in Figure 9 indicated that the surfaces of AgNPs produced by peel extracts (pomegranate and orange) had a negative charge of approximately −16.9 and −19.5 mV, respectively.

### 2.6. Antifungal Activity

In vitro, the antifungal potential of extracts of pomegranate and orange peels and their biosynthesized AgNPs are presented in Figure 10 and Figure 11. The peel extracts and the biosynthesized AgNPs have repressed the growth of *A. solani* mycelia at all concentrations used. The per cent inhibition of mycelial growth by the peel extracts and AgNPs varies among tested peels. The reduction in the diameter of the mycelial growth of *A. solani* was more by AgNPs than peel extracts or AgNO_3_. In addition, pomegranate peel extract and their AgNPs exhibited more reduction of mycelial growth than those of orange peel extracts. The maximum reduction in the mycelial growth by different treatments was verified at 100 µg/mL. This concentration exhibited the reduction of mycelial growth with pomegranate peel extract, their AgNPs, and AgNO_3_ by 6.12, 16.14 and 14.12 cm, respectively, while with orange peel extract, AgNPs and AgNO_3_ by 5.21, 12.52 and 10.61 cm, respectively.

### 2.7. Ultrastructural Studies

The treated hyphae of *A. solani* with the AgNPs biosynthesized from pomegranate peel extract were examined by scanning electron microscope (SEM) to give a clear image of how this extract could alter the fungal mycelia. Results established an obvious difference between the treated and untreated fungal mycelia. The untreated mycelia of *A. solani* were appeared as well-developed, inflated having a smooth wall (Figure 12A). Conversely, the treated mycelia with AgNPs at 100 µg/mL showed plasmolysis, distortion and squashing. Almost all hyphae were appeared empty, collapsed and completely dead (Figure 12B). In addition, the untreated and treated hyphae of *A. solani* were examined by TEM. The untreated hyphae show a typical cell wall, cell membrane and normal organelles (Figure 13A). Hyphae treated with AgNPs (100 µg/mL) biosynthesized from pomegranate peel extract showing disintegration and deterioration of cytoplasm, breakdown of the cell membrane and cell wall, and collapse of hyphae (Figure 13B).

## 3. Discussion

The chief aim of this work was to synthesize AgPNs by pomegranate and orange peel extracts using the lowest concentration of AgNO_3_ solution for controlling the fungal pathogen, *A. solani*, causing the early blight of tomato plants.

Many investigators in various countries [29,30,31,32,33] have tried the morphological and molecular characterization of *A. solani.* Moreover, in the present study, the molecular investigation confirmed the morphological characteristics of the pathogen isolates that were suspected to be *A. solani*. Therefore, morphological characterization provided an excellent tool for species identification but could not specifically identify the isolates to species level. Okayo et al. [34] noted that morphological classification of fungal species lacks accuracy but it is important in assisting the organization of the fungal isolates into groups permitting easier scrutiny by advanced approaches. Furthermore, morphological characteristics such as colony colour and texture, size and shape of the conidia have been used to differentiate *Alternaria* species [35]. This study exposed high morphological variability within *A. solani* isolates.

Many authors [36,37] have reported the high genetic diversity of *A. solani*. Chaerani and Voorrips [38] showed that genetic variation may happen among isolates got from different lesions of the same leaflet. According to Craven et al. [39], genotypic variation in *A. solani* is produced by the ability of its mycelia to communicate by bridges constructed through hyphal fusion that permit the distribution of nutrients, water and signalling molecules all over the colony. Genetic diversity is also provided by mutations, selection and gene flow [40], heterokaryosis that result from hyphal anastomosis, recombination and movement of the pathogen over prolonged expanses [41].

The crude extract of pomegranate and orange peels was analyzed using HPLC to detect the main phenolic components that could play a key role in the suppression of the tested pathogen. Moreover, results approved many phenolic compounds in the different extracts. These differences may be related to the fruit variety, the environmental conditions in which the fruits were cultivated and the antimicrobial properties of each extract. The presented results approved the occurrence of certain effective composites for example Quercitrin and Chlorogenic acid in pomegranate and orange peel extracts.

Phenolic compounds are aromatic benzene rings with one or additional hydroxyl groups made by plants mostly for protection against anti-stress [42]. Furthermore, phenolics are the major unit of secondary metabolites, which are extensively dispersed in most of the plants and show a significant function in plant resistance versus many diseases [43]. Furthermore, they act as defensive agents, and pesticides against fungal pathogens [44]. A quantity of simple and complex phenolics gather in plants and act as phytoalexins and phytoanticipins against plant pathogens [45]. Consequently, phenolic compounds have been suggested to work as useful substitutes to the chemical control of plant pathogens of agricultural crops [46]. Plants react to pathogens attack by gathering phytoalexins, such as hydroxycoumarins and hydroxycinnamate conjugates [47].

Since metallic NPs show their characteristic and distinctive absorbance peak because of surface plasmon character [48]. The development of AgNPs in the reaction medium was confirmed shortly after the synthesis through UV-vis spectroscopy. The UV-vis spectrum of fabricated AgNPs in aqueous colloidal solution revealed its maximum absorbance at 437 nm and 450 nm for pomegranate and orange peels, respectively, after 72 h, which is the expected characteristic surface plasmon of AgNPs [49].

Fourier transforms infrared analysis was applied to identify different functional groups of biomolecules available in pomegranate and orange peel extracts which played as reduced and stabilized nanoparticles by producing a coating layer on the surface of the NPs [50]. FT-IR spectra, therefore, suggested that fabricated AgNPs were attached by various active phytochemicals of pomegranate and orange peel extracts, which created a coating layer encapsulating the NPs to stabilize them and prevent aggregation and agglomeration [51].

In the present study, the AgNPs in the TEM image are dispersed as roughly as spherical with very good distributions. Besides, the particle size distribution and Zeta potential measurements of biosynthesized AgNPs using pomegranate and orange peel extracts were also considered. Particle dimension distribution in terms of Z-average value was consistent with TEM outcomes. Instead, the average zeta potential value that shows the surface charge of AgNPs in colloidal solution was determined. In a suspension, great negative or positive potential value established excellent physical consistency of colloidal NPs because of electrostatic repulsion of individual grains [52]. Besides, z-values larger than ±30 mV show the monodisperse nature of colloidal nano-suspension, whereas lower potentials, less than ±5 mV, signifies aggregation and agglomeration [52]. According to TEM and Zeta analyses, it can be fulfilled that biosynthesized AgNPs using lemon peel extract showed their polydisperse nature, which was distributed with the absence of aggregation and accumulation.

The current study revealed that AgNPs biosynthesized from pomegranate peel extract inhibit the fungal growth more than orange peel extract. This is maybe related to the variation in the variety of fruits. Furthermore, SEM and TEM examined the treated hyphae of *A. solani* with AgNPs biosynthesized from pomegranate peel extract. Results established an obvious difference between the treated and untreated mycelia. The treated hyphae with AgNPs at the concentration of 100 µL showed remarkable changes when compared with the control as observed by SEM. Hyphal cells treated with AgNPs showed a disintegration and deterioration of cytoplasm, breakdown of the cell membrane and cell wall as observed by TEM.

The applications of NPs as fungicides have been intensifying progressively in current years owing to several advantages over traditional chemical fungicides. The present study gives fungicidal efficacy of AgNPs against *A. solani*. El-Batal et al. [53] stated that AgNPs interrupt passage schemes comprising ion efflux. The disruption of ion efflux can generate a fast buildup of silver ions, stopping cellular movements at their inferior concentrations such as metabolism and respiration by responding with molecules. Moreover, Ag++ are distinguished to create reactive oxygen types through their reaction with oxygen, which is harmful to cells, producing injury to lipids, proteins and nucleic acids. Furthermore, diverse constraints, such as fragment size and concentration, which affect the competence of the fungicidal influence, presented that small-sized particles (12.7 nm) are very active in impeding fungal development. It is known that AgNPs are very deadly to most microorganisms. Several studies suggest that AgNPs may connect to the surface of the cell membrane interrupting permeability and respiratory purpose of the cell. It is likely that AgNPs not only interact with the membrane surface but can also enter within the microorganism [54]. Additionally, silver impedes the metabolism of pathogen and the electrical load of an emitted silver anion kill the pathogens by governing a proliferation function [55].

Moreover, same results were documented by Gajbhiye et al. [56] who detected that, the TEM investigation discovered the collaboration between nano-silver and the membrane construction of *Candida albicans* cells through nano-silver disclosure led to deviations in the membranes of this fungus, which could be differentiated as holes on the membrane surface and later leads to cell killing. In this process, the AgNPs dispersed gradually as Ag ions. Moreover, the ions replace the -SH bonds to -SAg ones in the microorganism’s membrane in a substitution reaction, leading to the fatality of the microorganism [57]. Guilger-Casagrande and de Lima [58] examined synthesizing of highly stabilized the monodispersed AgNPs by different fungi and the antifungal efficiency of AgNPs against various agricultural parasites. Besides, Lamsal et al. [59] reported that AgNPs initiated a harmful impact on the growth of *Colletotrichum* sp. as studied in laboratory and field. The molecular method offered with AgNPs can be certified to create free radicals. The free radicals provided oxidative tension and cell death in most microorganisms by affecting the organelles inside the cells of microorganisms [60].

## 4. Materials and Methods

### 4.1. Isolation of the Fungal Pathogen

Freshly infected diseased leaves shows that typical symptoms of early blight were collected from various areas in Saudi Arabia. The infected leaves showing early blight symptoms were sliced into minute pieces measuring about 5 mm, their surfaces were sterilized by 1% NaOCl solution up to 1 min, and next they were washed by sanitized DW. The slices were put on PDA medium and incubated for 12 h light and the same in dark at 25 °C. Pure fungal culture was gained by the isolation method of hyphal tip.

### 4.2. Pathogenicity Test

The pathogenicity test of all *Alternaria* isolates was confirmed by Koch’s postulates. The seeds of tomato (cultivar Super Strain B) of the same size were surface sterilized in 2% NaOCl for 5 min and washed three times with distilled water. The fungal inoculum was planned by culturing every tested isolate on PDA medium at 27 °C for 15 days. Afterwards, 10 mL of sterile DW was inserted into every Petri dish, and the colonies were gently rubbed with a disinfected needle. The resultant conidial suspension from each isolate was corrected to 5 × 10^6^ spores per ml and designed for the inoculation of 20 tomato plants with an atomizer. Following inoculation, plants were shielded with plastic bags for 48 h to keep high humidity. Subsequent to 48 h, the plastic bags were detached and plants were sheltered in pots under conditions of a greenhouse. Two weeks after inoculation, the results were compared, and the experiment was replicated twice.

### 4.3. Identification of the Fungal Pathogen

#### 4.3.1. Morphological Features

The fungal isolates were identified by a light microscope at a magnification of 40×, including an examination of the size, structure and shape of the conidia. The isolates were identified according to the criteria for cultural and morphological characteristics explained by Naik et al. [29].

#### 4.3.2. Molecular Identification of Alternaria isolates

##### Extraction of DNA

Extraction of DNA was done by acetyl trimethylammonium bromide (CTAB). The fungal isolates were cultured using a PDA medium for 7 days at 25 °C. Next, the fungal mycelia were accumulated with a disinfected scalpel, moved to a sterilized 1.5 mL Eppendorf micro-centrifuge and crushed using liquid nitrogen reaching a consistent powder. Subsequently, 700 μL of CTAB buffer (100 mM Tris-HCl, 1.0 M NaCl, 10 mM EDTA, pH 8.0) containing CTAB (2%, w:v) was added to 100 g of the crushed powder. After mixing, the microtubes were positioned at 65 °C for 45 min and then centrifuged for 10 min at 10,000× *g*. Consequently, 650 μL of the supernatant mixture, besides an equal volume of chloroform and isoamyl alcohol, was centrifuged at 10,000× *g* for 10 min at room temperature. The supernatant was separated, and 0.7 mL of cold isopropyl alcohol was inserted into the combination and stored at −20 °C for 20 min. Afterwards, the tubes were subjected to centrifugation for 5 min at 10,000× *g*. The precipitate comprising DNA was washed twice with 70% ethyl alcohol and centrifuged at 10,000× *g* for 5 min every time. Ultimately, the pellet was dried in the air, and the prepared DNA was resuspended in 30 μL of TE buffer (10 mM Tris-HCl pH of 8, 1 mM EDTA). Nucleic acid concentrations were controlled by a NanoDrop 1000 Spectrophotometer (Thermo Fisher Scientific, Wilmington, DE, USA). Furthermore, the reliability of each DNA sample was examined on 12 g L^−1^ agarose gel. Besides, the property of the extracted DNA on 1% agarose gel was considered with DAC gel by using electrophoresis and DNA safe staining.

##### Polymerase Chain Reaction

The amplification of the ITS1 area in PCR was achieved by two primers, viz. ITS1 (5′-TCCGTAGGTGAACCTGCGG-3′) and ITS4 (TCCT CCGCTTATTGATATGC-3′) [61]. The PCR reaction was attained in a Biorad thermocycler (S 1000TM). The PCR comprised a unique denaturation step of 5 min at 94 °C with 40 cycles for 1 min at 94 °C, primer hardening for 45 s at 53 °C, and primer expansion for 90 s at 72 °C through a primary denaturation at 94 °C for 5 min and a latter extension for 10 min at 72 °C. The reaction was done in a volume of 25 μL comprising 2 μL DNA templates, 12.5 μL master mix and 10 pmol of every primer. The PCR outcomes were electrophoresed for 1 h at 80 V in 0.8% agarose gel in trisborat EDTA buffer at pH 8. The gels were stained with DNA safe stain (10 mg/mL) and viewed in a gel documentation system (Alpha Innotech, San Leandro, CA, USA). Next, the products gained from the PCR of the ITS1 region were sequenced. Moreover, the nucleotide sequences achieved by the local BLAST (http://blast.ncbi.nlm.nih.gov/Blast.cgi) (13 July 2021) were tested and contrasted with parallel sequences in the Gene Bank.

### 4.4. Collection and Drying of Selected Fruit Peels

Two types of fresh fruits, for example, *Punica granatum* L. var. mangulati (pomegranate and *Citrus sinensis* L. var. baladi (orange) were collected from local markets in Saudi Arabia. The fruits were peeled with a sharp knife, washed by DW and then dried in the air and placed in the shade at room temperature for 3–4 weeks before grinding by an electrical blender for 3 min.

### 4.5. Preparation of Peel Extracts

For extraction, precisely 100 g powder of each sample was soaked in 1 L of deionized water for 48 h at room temperature and preserved in the dark. The blends were first filtered with Whatman No. 1 filter paper and centrifuged at 9660× *g* for 30 min at 4 °C. The extract was intensed with a rotary evaporator at 60 °C and then dried in an oven at 50 °C for 48 h [8].

### 4.6. Determination of Total Phenolic Compounds

The phenolic compounds concentration in peel extracts was demonstrated under the method described by Jayaprakasha et al. [62]. The data were expressed as tannic acid equals. Two hundred ml of extracts were liquefied in a combination of methanol and water (6:4 *v*/*v*). Sample of 0.2 mL was combined by 0.1 mL of tenfold-diluted Folin-Ciocalteu reagents and 0.8 mL of 7.5% sodium carbonate solution. After settling for 30 min at room temperature, the absorbance was calculated at 765 nm by a Spectrophotometer.

### 4.7. Determination of Phenolic Compounds by HPLC

The separation of phenolic components from fruit peel extracts dissolved in methanol was achieved by HPLC system (Agilent 1200 Series) organized with an opposite phase C18 column (150–4.6 mm, 0.5 mm in particle size), a quaternary pump and a UV sensor set at 330 nm. A two-descent elution at a flow rate of 1 mL/min. The mobile phase included acetonitrile (A) and methanol (B). The parting of phenolic compounds commenced with a linear gradient of 20% B (0–10 min), 40% B (10–25 min), 90% B (25–30 min). The temperature of the column was preserved at 35 °C and the injection volume was 20 mL. The recognition of phenolic components was done on retention time and UV spectra with existing standards. The quantity of phenolic components was accomplished through the peak zones verified at 330 nm. Results were conveyed as mg per gram peel powder.

### 4.8. Biosynthesis of Silver Nanoparticles (AgNPs)

For AgNPs green synthesis, 3 mL of every fruit peel extract was cautiously mixed with 40 mL of 1mM aqueous AgNO_3_ solution in a test tube and reserved at 25 °C for 5 h apart from light. The alteration of the color of colloidal suspension from yellow to dark brown and from bright red to dark brown proved the synthesis of AgNPs [48].

### 4.9. Characterization of Silver Nanoparticles

The classification of AgNPs was prepared through an observation of a change in color using UV-vis Spectrophotometer (Beckman DU-40). The biosynthesized AgNPs was permitted by selecting of the reaction combination at steady periods and the absorption spectra were visualized at the wavelength of 370–750 nm by Unicam UV-vis Spectrometer UV2, USA. FTIR Spectrometer (Perkin-Elmer LS-55-Luminescence Spectrometer) reviewed the chemical conformation of the synthesized AgPNs. The solutions were dried to powder at 75 °C and these powders were calculated in the range between 4000–400 cm^−1^ by KBr pellet method. AgPNs were analyzed using TEM a speeding up voltage of 200 kv using TEM JEOL JEM-2100. In addition, Zeta potential studies and size distribution by volume, the nanocolloidal solution constancy and surface charge of NPs were considered using Zeta Potential Analyzer (Model Malvern Zeta-Size Nano-zs90, USA).

### 4.10. Antifungal Activity of Crud Peel Extracts and Their Biosynthesized AgNPs

The antifungal activity of peel extracts of pomegranate, orange and their biosynthesised AgNPs was examined according to Singh et al. [63]. Five-millilitre peel extracts and their biosynthesised AgNPs at diverse concentrations (20, 40, 60, 80 and 100 µg/mL) were inserted into 5 mL of the sterilized media before solidification. The combination was moved into 9-cm Petri dishes. Negative controls including media only were also measured. Petri dishes were next incubated apart from the light at 25 °C for 48 h. Afterwards, the Petri dishes were inoculated by 5 mm in diameter fungal plugs. Then, the plates were incubated in the dark at 25 °C for additional five days. The radial growth of fungal mycelia was assessed as the mean of diameters of two fungal colonies. The inhibition effectiveness of crude peel extracts and their AgNPs about the fungal pathogen was determined by the next equation: Linear growth reduction (percentage) = control-treatment/control × 100.

### 4.11. Ultrastructural Studies

Since AgNPs biosynthesized from pomegranate fruit peels (at 100 µg/mL) exhibited the greatest antifungal activity of *A. solani*, both SEM and TEM examined its culture. For SEM planning, fungal plugs (1 mm^3^) were fixed in 3% glutaraldehyde in 0.1 M sodium cacodylate buffer at pH 6.8, dehydrated in ordered sequences of acetone and dried in a critical point dryer (Polaron CPD 7501, Watford, UK). The samples were coated with gold-palladium with JFC-1600 Auto fine coater (Polaron SC7620, UK). Fungal mycelia were then examined and photographed under Jeol SEM-6400JSM-6360LV (JEOL Ltd., Akishima, Japan). In the case of TEM, the hyphal masses of the pathogen were prefixed in 3% glutaraldehyde in 0.1 M sodium cacodylate buffer at pH 6.8; post-fixed in 1% OsO_4_; dehydrated in a graded series of ethanol; embedded in Araldite resin and sectioned by using an Ultra-microtome. Ultrathin sections were stained with uranyl acetate and then by lead citrate and observed by JOEL model JEM-1230 TEM (JEOL Ltd., Akishima, Tokyo 196-8558, Japan).

### 4.12. Statistical Study

The obtained results were analyzed by Costas software (version 6.4), for a one-way much-randomized block scheme. Duncan’s multiple range investigations were employed to compare means at a probability (*p*) level of ≤0.01.

## 5. Conclusions

This study established a rapid, simple, inexpensive and environment-friendly approach to the construction of silver nanoparticles using the extracts of pomegranate and orange peels. The decline of ionic Ag to their nanoparticles and succeeding capping for stabilizing nano-silvers was suggested to happen throughout the participation of various active phytochemicals of these extracts. Moreover, the AgNPs formed in this study showed their polydisperse nature and confirmed excellent stability for several months with the absence of aggregation and accumulation. In vitro, these AgNPs biosynthesized from pomegranate and orange peel extracts were succeeded to prevent the growth of *A. solani*. Furthermore, the non-toxic nature of these AgNPs at the concentration may expand their significance for further studies related to medical science, healthcare, veterinary medicine, cosmetics, food industry and nanobiotechnology.

## Figures and Tables

**Figure 1 plants-10-02363-f001:**
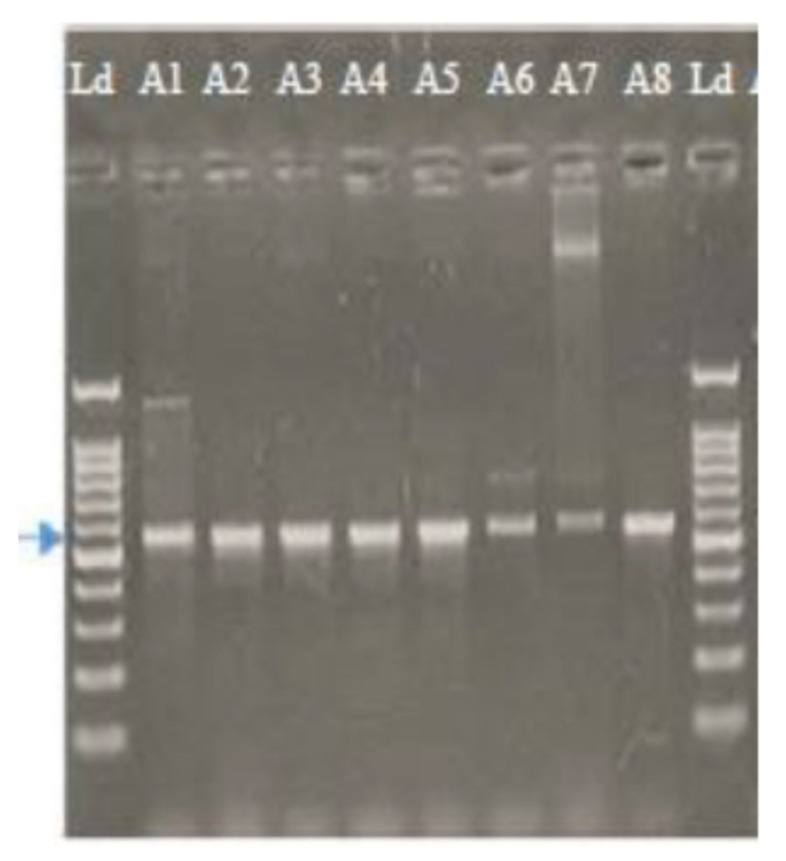
Gel image showing 570 bp (arrowhead) DNA fragment amplified by ITS (1 and 4) primer in *A. solani* isolates on 1.2% agarose gel electrophoresis. Ld is the l kb RTU 1151021805 ladder.

**Figure 2 plants-10-02363-f002:**
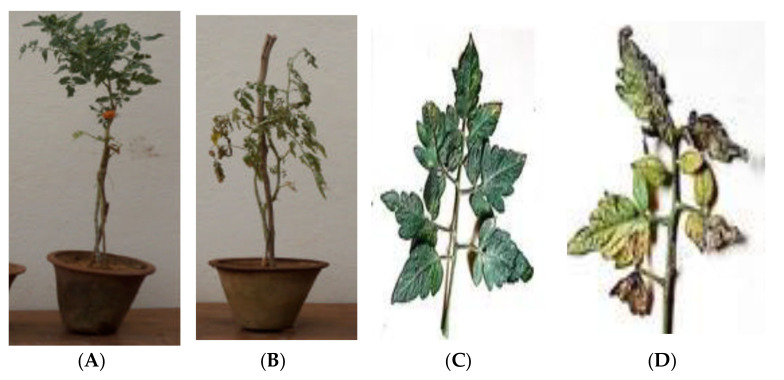
Pathogenicity test of tomato plant by *A. solani* (Isolate A1) under greenhouse conditions. (**A**). Healthy control in a pot. (**B**) Infected in a pot. (**C**). Healthy control as a single branch of tomato plant. (**D**). Infected branch of tomato plant.

**Figure 3 plants-10-02363-f003:**
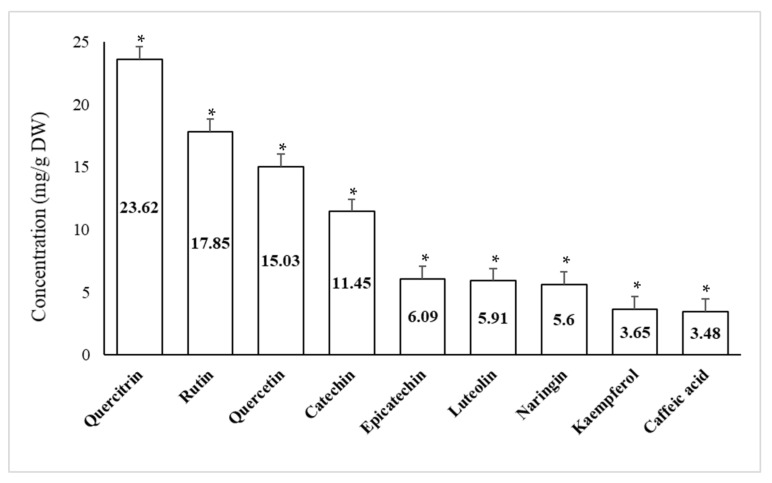
Common phenolic compounds of pomegranate peel powder (Highly significant = * *p* < 0.01, n = 3).

**Figure 4 plants-10-02363-f004:**
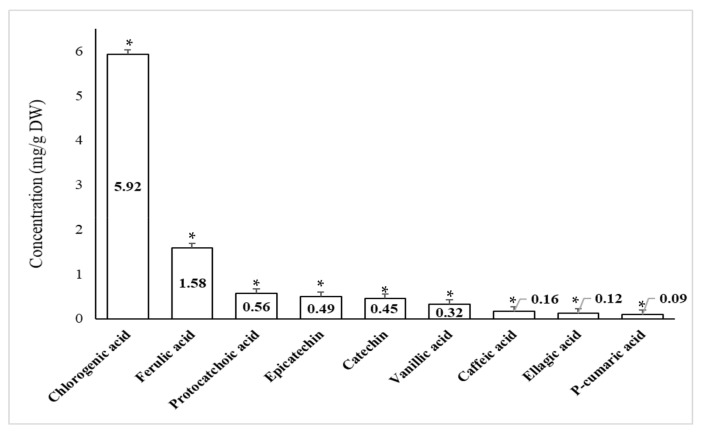
Common phenolic compounds of orange peel powder (Highly significant = * *p* < 0.01, n = 3).

**Figure 5 plants-10-02363-f005:**
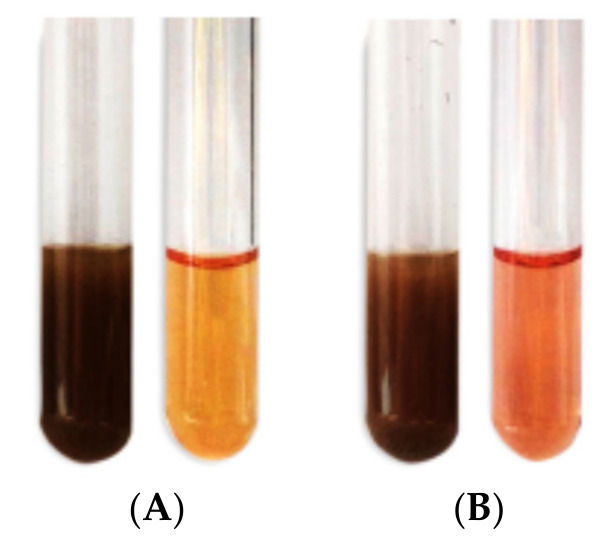
Visual representation of biosynthesis of AgNPs from pomegranate peel extract (**A**) and orange peel extract (**B**).

**Figure 6 plants-10-02363-f006:**
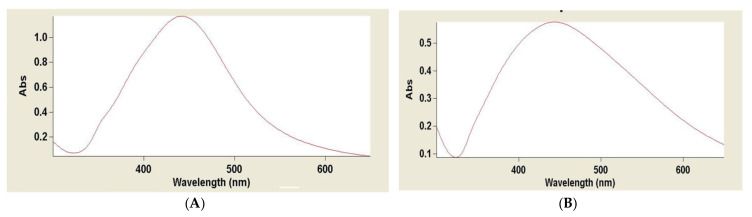
The UV-Vis spectra recorded for the reaction of Pomegranate peel (**A**) and Orange peel (**B**) with AgNO_3_ solution.

**Figure 7 plants-10-02363-f007:**
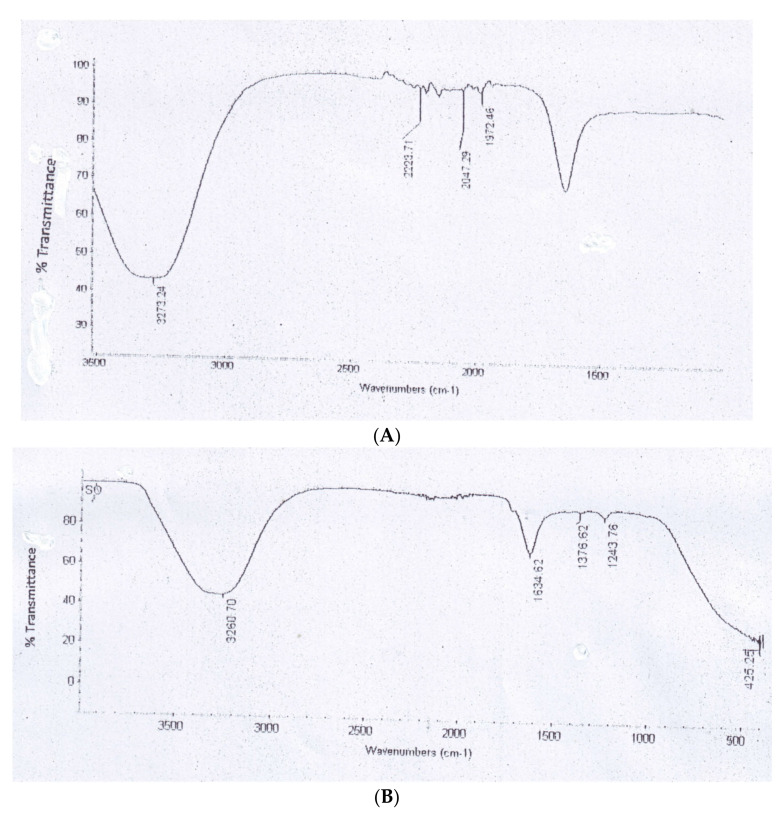
FTIR adsorption spectra of AgNPs prepared by pomegranate peel (**A**) and orange peel (**B**).

**Figure 8 plants-10-02363-f008:**
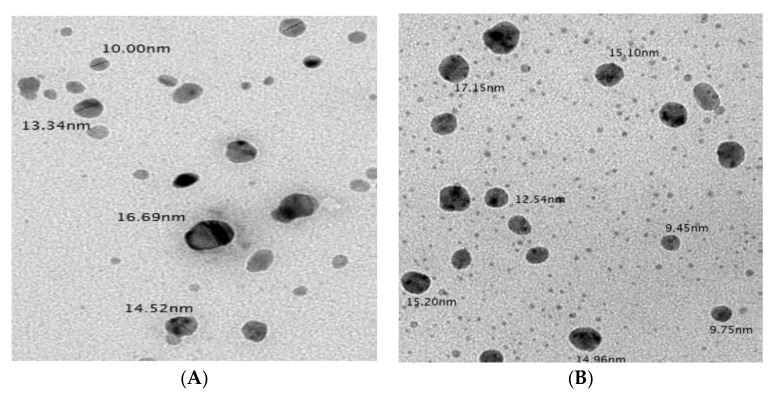
TEM micrographs of optimized biosynthesized AgNPs showing a spherical shape of nanocolloids and their size distribution ranges. (**A**) AgNPs biosynthesized using pomegranate peel extract (**B**) AgNPs biosynthesized using orange peel extract. Bar = 100 nm.

**Figure 9 plants-10-02363-f009:**
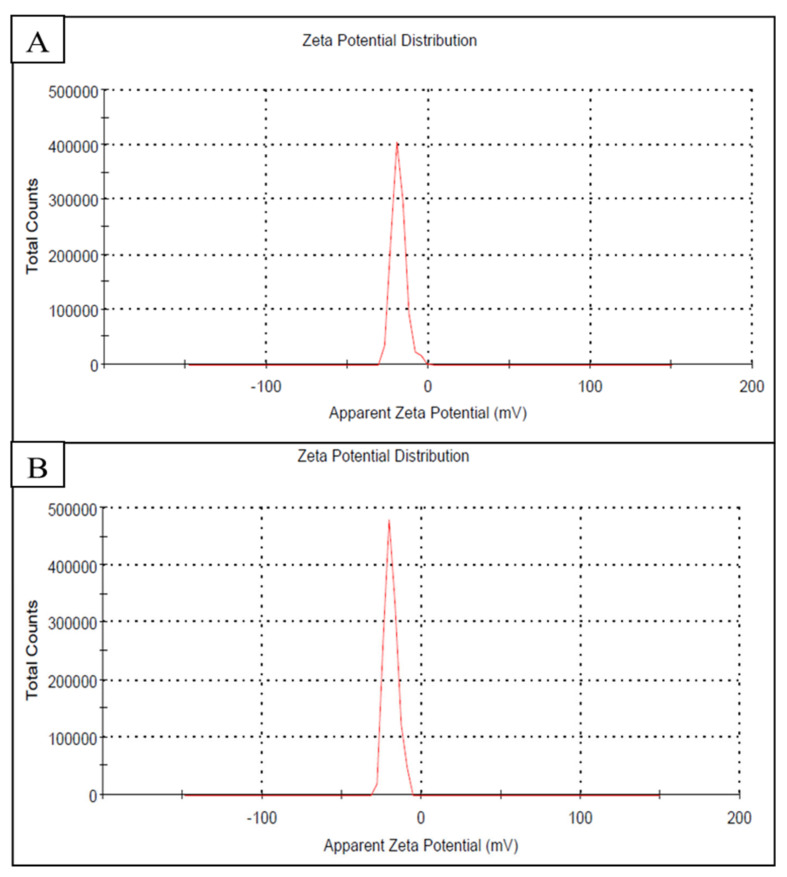
(**A**) Zeta potential of AgNPs synthesis by pomegranate peel showed −16.9 mV. (**B**) Zeta potential of AgNPs synthesis by orange peel showed −19.5 mV.

**Figure 10 plants-10-02363-f010:**
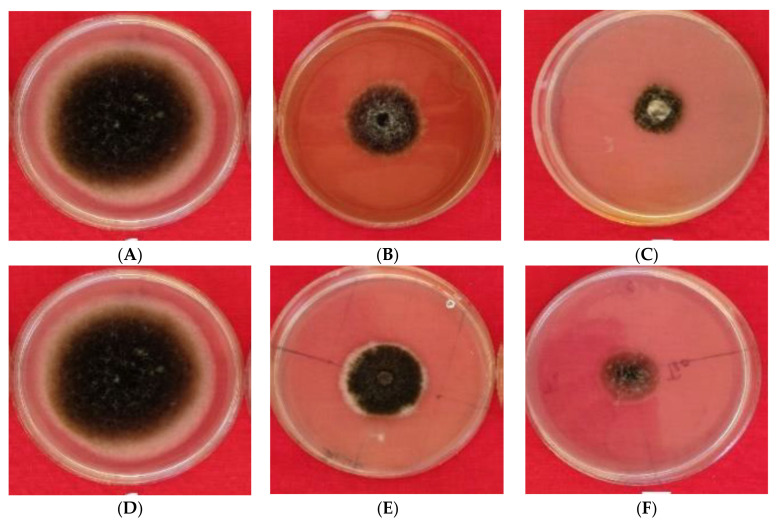
Effect of different fruit peel extracts and their AgNPs on radial growth of *A. solani* (cm) at the concentration of 100 µg/mL. Control, medium only (**A**,**D**), pomegranate peel extract (**B**) and their AgNPs (**C**), orange peel extract (**E**) and their AgNPs (**F**).

**Figure 11 plants-10-02363-f011:**
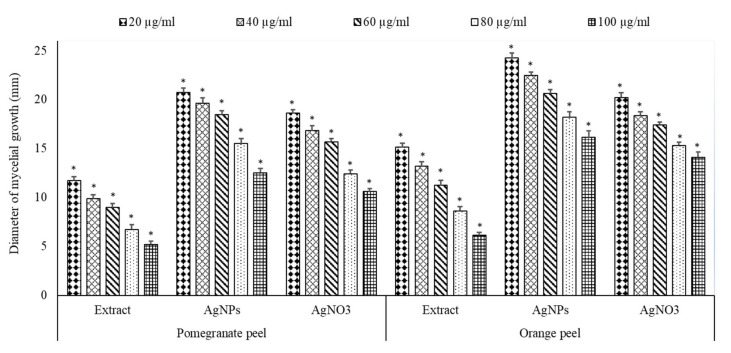
Antifungal activity of crude extract of pomegranate and orange peels, AgNPs and AgNO_3_ against *A. solani* (Highly significant = * *p* < 0.01, n = 3).

**Figure 12 plants-10-02363-f012:**
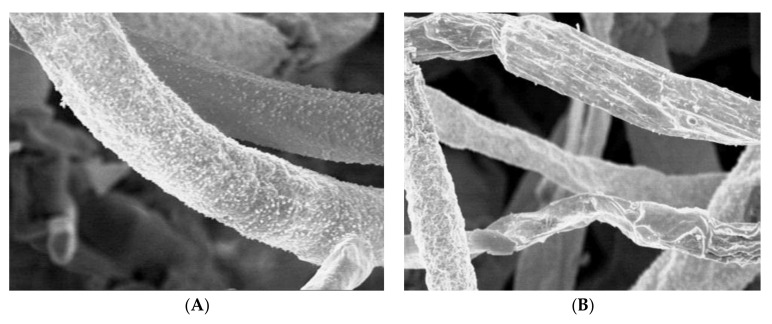
SEM micrographs of *A. solani*. (**A**). The untreated mycelia are well-developed inflated having normal wall. (**B**). The treated mycelia by AgNPs (100 µg/mL) showing plasmolysis, distorted, squashed and collapsed hyphae and completely flat and empty dead hyphae. Scale bar = 5.0 µm.

**Figure 13 plants-10-02363-f013:**
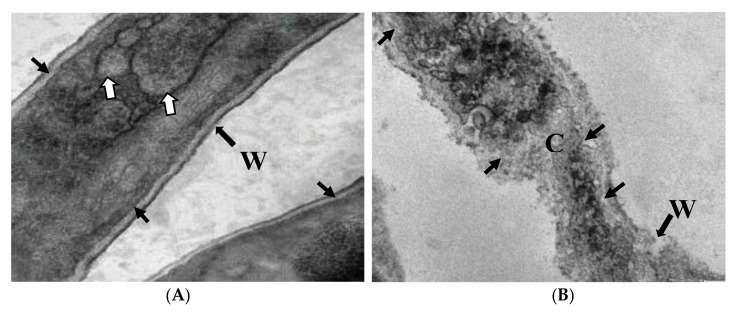
TEM studies of a longitudinal section of *A. solani* hypha. (**A**). Normal untreated hypha showing typical cell wall (W), cell membrane (arrow) and organelles (short arrows). (**B**). Hyphae treated with AgNPs (100 µg/mL) biosynthesized from pomegranate peel extract showing disintegration and deterioration of cytoplasm (CY), break down of the cell membrane (arrow) and cell wall (W) and collapse of hyphae. Scale bar = 0.5 μm.

**Table 1 plants-10-02363-t001:** Morphological variability of *Alternaria solani* isolates.

Groups	Isolate Code	Colony Color	Substrate Color	Margin Color	Margin Growth	Transverse Conidial Septa	Longitudinal Conidial Septa	Beak Elongation	Beak Branshing
1	A1	Dark brown	Dark grey	Brownish white	Circular	Three	One	Slender short	Branched
2	A2, A3, A8	Greenish brown	Dark brown	Grey	Irregular	Three	None	Elongated	Unbranched
3	A4, A6	Dark grey	Greyish brown	White	Circular	Two	One	Elongated	Unbranched
4	A5, A7	Grey	Brown	Greyish white	Circular	Four	One	Selender short	Unbranched

**Table 2 plants-10-02363-t002:** Molecular variability of *A. solani* isolates.

Groups	Isolate Code	Accession Number	Closest Match	Similarity to Genebank Accessions	Frequency (%)
1	A1	MN871620	Clone 107	100	12.25
2	A2, A3, A8	MN871620	Clone 107	100	38.75
3	A4, A6	MN871615	Clone 180	99.50	24.50
4	A5, A7	MN871610	Clone 42	99.85	24.50

**Table 3 plants-10-02363-t003:** Total phenolic content (mg/g DW) in pomegranate and orange peels extracted by methanol.

Peels	Total Phenolic Content (mg/g DW)
Pomegranate	105.4 ± 1.98
Orange	86.32 ± 1.78

Each value is represented as mean ± SD (n = 3).

**Table 4 plants-10-02363-t004:** Average particle size and zeta potential values measured with a zetasizer nano.

Fruit Peel	Size (nm)	Zeta Potential (mV)
Pomegranate	8	−16.9
Orange	14	−19.5

## Data Availability

Not applicable.

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
