# Peer review of "Green Synthesis of Silver Nanoparticles Using Pomegranate and Orange Peel Extracts and Their Antifungal Activity against Alternaria solani, the Causal Agent of Early Blight Disease of Tomato"

_plants, 2021, doi:10.3390/plants10112363_

Round 1

Reviewer 1 Report

Lines 366-371: I think an image suggestive for the affected plants should be included.

Lines 378-379: please specify how the correction of fungal burden was made.

Lines 421-422: include details about sequencing (equipment, conditions etc.)

Lines 126-137: Were the sequences of your Alternaria solani isolates deposited in GenBank database? If yes, please provide their accesion numbers.

Lines 477-487: Which solvent has been used for AgNPs and peel extracts? It seems that it was not included into the control plates. Also, the dilution of culture medium (1:1 with extracts and AgNPs suspensions) could affect the fungal growth comparing with control plates where only culture medium was used.

Line 492: The concentration of 10 mg/ml has been tested or not? (see line 479).

Author Response

Dear Reviewer;

thank you for you valuable comments

Reviewer 2 Report

In its assumptions, the study is justified and interesting. Similar research has not been done so far. However, I have a few technical notes:

line 148 - point 2.3 correct the font size and remove the period

line 152 - table 3 incorrect value 8.6.32

line 185 - figure 6 to improve the quality of photos

line 190 - figure 7 I suggest that you add more detailed photos

line 259 - remove bold

line 528, 540, 562, 566, 592, 603, 640, 702 - remove the underline 

Author Response

Dear Reviewer

Thank you for valuable comments

Round 2

Reviewer 1 Report

The authors have extensively modified their manuscript and improved its quality. I consider that it is now publishable.